# Abnormal birth weight in urban Nigeria: An examination of related factors

**Olufunke Fayehun**[1]*, **Soladoye Asa**[2]

**1** Department of Sociology, University of Ibadan, Ibadan, Nigeria, **2** Department of Demography and Social Statistics, Obafemi Awolowo University, Ife, Nigeria

* cl_funke@yahoo.com

**Data Availability Statement:** We received approval for use of the Nigeria Demographic Health Survey dataset for this registered topic from the DHS program of ICF, Inc. United States. The procedure for public use of the DHS dataset does not permit the transfer of the dataset to others without the

## Abstract

There is a knowledge gap on abnormal birth weight in urban Nigeria where specific community contexts can have a significant impact on a child's health. Abnormal birth weight, classified into low birth weight and high birth weight, is often associated with adverse health outcomes and a leading risk for neonatal morbidity and mortality. The study used datasets from the birth recode file of 2013 and 2018 Nigeria Demographic and Health Survey (NDHS); a weighted sample of pooled 9,244 live births by 7,951 mothers within ten years (2008–2018) in urban Nigeria. The effects of individual, healthcare utilization and community-level variables on the two abnormal birth weight categories were explored with a multinomial logistic regression models using normal birth weight as a reference group. In urban Nigeria, the overall prevalence of ABW was 18.3%; high birth weight accounted for the majority (10.7%) of infants who were outside the normal birth weight range. Predictors of LBW were community (region), child characteristic (the type of birth) and household (wealth index) while that of HBW were community (regions), child characteristics (birth intervals and sex), maternal characteristic (education) and healthcare utilization (ANC registration). LBW was significantly more prevalent in the northern part while HBW was more common in the southern part of urban Nigeria. This pattern conforms to the expected north-south dichotomy in health indicators and outcomes. These differences can be linked to suggested variation in regional exposure to urbanization in Nigeria.

## Introduction

Urbanization has been linked to improved child health outcomes because of the assumed advantage in access to improved health care services, higher education and its associated benefits and good infrastructure [1]. However, rapid urbanization in megacities in Africa, as well as adaptation to the new urban culture and lifestyles [2–6] can affect the maternal quality of life during pregnancy and constitute significant predictors of health outcomes [1, 7, 8]. Exploring health outcomes such as birth weight in the urban setting is pertinent in the context of high rates of urban natural increase and changing lifestyle, which is due to the youthful age structure of urban population in Sub-Saharan Africa [9, 10].

Birth weight is a strong predictor of child's health and development [11–13] A low birth weight (LBW) of less than 2,500g is associated with adverse health outcomes and is a leading

written consent of the DHS program of ICF, Inc. Other researchers may apply for access from ICF Macro Institutional Data Access via the following instructions: https://www.dhsprogram.com/data/Access-Instructions.cfm.

**Funding:** OA: This research was supported by the Consortium for Advanced Research Training in Africa (CARTA). CARTA is jointly led by the African Population and Health Research Center and the University of the Witwatersrand and funded by the DAAD, Carnegie Corporation of New York (Grant No–B 8606.R02), Sida (Grant No:54100029) and the DELTAS Africa Initiative (Grant No: 107768/Z/15/Z). The DELTAS Africa Initiative is an independent funding scheme of the African Academy of Sciences (AAS)'s Alliance for Accelerating Excellence in Science in Africa (AESA) and supported by the New Partnership for Africa's Development Planning and Coordinating Agency (NEPAD Agency) with funding from the Wellcome Trust (UK) and the UK government. The statements made and views expressed are solely the responsibility of the Fellow.

**Competing interests:** The authors have declared no competing interests exist.

risk factor for neonatal morbidity and mortality [14–19]. Also, high birth weight (HBW) or macrosomia (more than 4,000g) has been associated with negative maternal and neonatal outcomes, complicated delivery and epidemics of obesity with its associated problems that span through childhood, adolescence and adulthood [20–27]. These two birth weight categories, often studied in comparison to normal birth weight (NBW), are termed abnormal birth weight (ABW) [28–30].

Although there is evidence of increasing birth weight in some developing countries [25], ABW is an under-researched child health topic in urban Africa. Most studies on birth weight in Africa have focused on low birth weight, with risk factors including maternal and child characteristics, healthcare utilization, household, nutrition, environment and community variables [14, 31–36]. On the contrary, studies on high birth weight in Africa have been few, but available evidence shows that the significant risk factors are maternal obesity, socioeconomic condition, residence and gestational diabetes [22, 37–41]. Moreover, only a very few studies [15, 36, 42] explored the pattern and predictors of both low and high birth weight in the context of urbanization [43, 44] in Africa.

Even in Nigeria, the largest African population with rapid urbanization and most complex urban system [9, 10], few recent studies [39, 40, 45–47] explored the risk factors of both LBW and HBW. Besides, there is a general low reporting of birth weight in Nigeria, although the Nigeria Demographic and Health Survey (NDHS) 2013 and 2018 estimated birth weight based on numerical values from written records and mother reports [48, 49]. Based on that estimation, only 16 percent and 24 percent reported birth weight in NDHS 2013 and 2018 respectively. The prevalence rate of LBW in 2018 is 7 percent, with rural (6.9 percent)-urban (7.5 percent) differentials [48]. While there was no national estimate of HBW, few hospital-based studies [26, 27, 39, 40, 50] reported prevalence of HBW range of 2.5 percent to 14.5 percent in different parts of Nigeria. In addition to the poor nationwide summary estimate of ABW, there is a knowledge gap on the predictors of ABW in Nigeria, particularly in urban areas where specific community contexts can have a significant impact on birth weight and child health.

This study, therefore, examines abnormal birth weight categories-LBW and HBW- in urban settings in Nigeria. The main questions guiding the study include: what is the prevalence of ABW (LBW and HBW) in urban Nigeria? What are the odds of ABW in urban Nigeria at the community and individual levels? These questions will be answered by exploring community and individual contexts of abnormal birth weight (ABW) in urban Nigeria using a nationally representative NDHS 2013 and 2018 dataset which covers records of livebirths from 2008 to 2018.

## Materials and methods

The study used datasets from the birth recode files of 2013 and 2018 Nigeria Demographic and Health Survey (NDHS). The two NDHS data sets are the fifth and sixth in the series of nationally representative surveys that collect information on basic demographic and health indicators in Nigeria. Detailed survey methodology, including sample design and data collection procedure, was published in the NDHS Reports for the survey years [48, 49]. The Nigeria birth recode file was downloaded after obtaining permission from ICF Inc. USA to use it for this study.

The birth recode file contains birth-related information of live births obtained from eligible women age 15 to 49 years in the 36 states and the Federal Capital Territory. The analytical sample for this study was based on (i) live births in the last five years preceding the survey (ii) availability of numerical values for the birth weight from written record or mothers report (iii)

urban residence. There were 3818 and 5604 live births in the five years preceding the survey with reported numerical values for weight at delivery in the urban areas of Nigeria for NDHS 2013 and 2018 respectively. This gives a weighted sample of pooled 9,244 livebirths by 7,951 mothers within ten years (2008–2018) in urban Nigeria.

## Outcome and exposure variables

Birth weight, measured in grams, is the outcome variable in this study. Information on birth weight was obtained from written records or mothers report. The continuous variable was categorized into low birth weight (LBW = less than 2500), normal birth weight (NBW = 2500g-4000g) and high birth weight (HBW = above 4000g). Further classification combined LBW and HBW to estimate the abnormal birthweight (<2500g and >4000g).

The exposure variables were grouped into individual, healthcare utilization and community levels. The community variable, region, was based on geopolitical delineations in the country and categorized into North Central, North East, North West, South East, South South and South West. Individual-level variables consist of infant, and maternal characteristics were birth order, sex of the child, preceding birth interval, maternal age at delivery, maternal education. (i) Type of birth was classified into singleton and multiple; (ii) Preceding birth interval grouped into less than 24 months and 24 months and above; (iii) Sex of child as male and female; (iv) Birth order was grouped into two: 1–3 and 4+; (v) Maternal Body Mass Index was classified into underweight ($<18.5 kg/m^2$); normal weight ($18.5–24.9 kg/m^2$), overweight and obese ($25.0 kg/m^2$ and above), (vi) Maternal age at delivery was computed as the difference between mother and baby's date of birth and grouped into below 20 years, 20-34years and 35-49years; (vii) Maternal education was classified into no formal education, primary education and secondary/higher education; (viii) Wealth index was regrouped into poor, middle and rich.

Healthcare utilization was examined because of the assumed proximity of health care facilities to urban residents, which predispose them to access proper care during pregnancy and its associated outcome. Antenatal care utilization was measured in terms of early ANC registration, more than 4 ANC visits and prenatal care by a skilled healthcare worker. This was assessed as (i) ANC visit, defined as the number of ANC visits by the mother of the index birth, categorized into two—below 4 visits, and 4 visits or more; (ii) ANC registration, defined as the month of first ANC registration by mothers of the index birth, grouped into the first trimester and after 1st trimester; (iii) Prenatal care by a skilled provider was defined as births whose mothers received antenatal care from a skilled healthcare provider. Skilled healthcare workers (SHW) in NDHS are doctor, nurse/midwife and auxiliary nurse/midwife. This was grouped into yes or no.

## Statistical analysis

The statistical analysis was at bivariable and multivariable levels. At the bivariable level, cross-tabulation and Chi-Square test of association between birth weight and selected factors at the individual, healthcare and community levels was examined. This showed the percentages of NBW, LBW and HBW in urban Nigeria between 2008 and 2018. Also, the yearly distribution of NBW, LBW and HBW was presented to show the pattern over the ten years covered by NDHS 2013 and 2018.

The effects of individual, healthcare utilization and community-level variables on the two abnormal birth weight categories were explored with a multinomial logistic regression model using normal birth weight as a reference group. This is to predict the odds of low and high weights in urban Nigeria. Model 1 has community-level variable (region) to show the odds of

low and high birth weights in urban Nigeria. Model 2 included individual-level characteristics of the child and mother (type of births, birth interval, sex of the child, birth order, maternal BMI, maternal age at delivery and maternal education, wealth index) and health care utilization variables. The risk of low and high birth weights in urban Nigeria was presented as odds ratio with a 95% confidence interval.

### Ethical clearance

Ethical clearance to conduct Nigeria Demographic and Health Surveys was obtained from National Health Research Ethics Committee of Nigeria (NHREC) and the ICF Institutional Review Board., United States. NDHS data are public access data and permission were granted to download the dataset for this study by Demographic and Health Survey Program of ICF Macro.

### Results

The percentage distribution of birth weights by background characteristics is presented in Table 1. Overall, 81.7% of live births in urban Nigeria between 2008 and 2018 reported normal birth weight, between 2,500g and 4,000g. The prevalence of HBW, estimated to be 10.7%, was higher than the LBW estimated at 7.5%. There was a significant association (p<0.05) between birth weight categories and all background characteristics except antenatal care from a skilled health care worker. The trend in abnormal birthweight among infants in urban Nigeria from 2013 to 2018 shows that there was a slight decrease in the estimated values for the two periods from 20.2 percent to 17 percent. Nearly one out of six live births in urban Nigeria reported abnormal birth weight for the period 2008–2018. Abnormal birth weight was highest for infants who were multiple births (32.7%), of mothers with no formal education (24.8%) and in the North West (22.7%). Other higher estimates of abnormal birth weight were among groups of infants with longer birth interval (18.6%), males (19.3%), higher birth order (19.5%), less than 4 ANC visits (21.4%), older mothers (18.9%), overweight/obese mothers (20.5%) and poor household (21.0%).

Table 2 shows the result of multinomial logistic regression for predictors of LBW and HBW in urban Nigeria. Although HBW and LBW are both abnormal birth weights, the predictors were different in the estimated models for urban Nigeria.

For the LBW Models, model 1 shows that community-level variable (geopolitical region) was significantly associated with LBW in urban Nigeria. The odds of LBW among infants in the North West (OR 3.91; 3.15–4.86), North East (OR 2.54; 95% CI 1.89–3.39), North Central (OR 1.35; 95% CI 1.01–1.80) and South South (OR 1.36; 95% CI 1.04–1.78) were significantly higher than the South West. South East has the lowest odds of LBW in Model 1 (OR 0.79; 95% CI 0.62–1.01), which was not significantly lower than South West. The inclusion of individual and healthcare utilization variables in Model 2 removed the significance of geopolitical region on LBW except North West which increased significantly by 64%. All the included variables for LBW in Model 2 were not significant except for singleton type of birth (OR 0.19; 95% CI 0.10–0.35), birth order of 1–3 (OR 0.72; 95% CI 0.52–0.99) and maternal body mass index for overweight/obese (OR 0.68; 95% CI 0.49–0.92). Multiple births had 81% higher risk of LBW than singleton births. Higher birth order of more than 3 has a higher risk of LBW than lower birth order.

For the HBW models, model 1 shows that odds of HBW were significantly lower in North West (OR 0.26; 0.18–0.40) and North East (OR 0.71; 95% CI 0.50–0.99) than South West. The odds of having infants with HBW were substantially higher in South South (OR 1.19; 95% CI 1.02–1.40) and South East (OR 1.41; 95% CI 1.16–1.72) than other regions. The regional

**Table 1. Percentage of normal and abnormal birth weights among different groups in urban Nigeria: NDHS 2008–2018.**

| Characteristics | Normal (NBW) | Abnormal Birth weight (ABW) | | | N |
| --- | --- | --- | --- | --- | --- |
| | | Low (LBW) | High (HBW) | LBW & HBW | |
| **Region*** | | | | | |
| North West | 77.3 | 20 | 2.7 | 22.7 | 938 |
| North East | 79.2 | 13.4 | 7.4 | 20.8 | 530 |
| North Central | 82.6 | 7.4 | 10 | 17.5 | 937 |
| South East | 82.7 | 4.3 | 12.9 | 17.3 | 2345 |
| South South | 78.4 | 7.1 | 14.5 | 21.6 | 1163 |
| South West | 83.6 | 5.5 | 10.9 | 16.5 | 3509 |
| **Type of Birth*** | | | | | |
| Singleton | 82.4 | 6.7 | 10.9 | 17.6 | 9037 |
| Multiple | 67.3 | 25.5 | 7.3 | 32.7 | 385 |
| **Birth Interval*** | | | | | |
| Less than 24 months | 83.9 | 6.9 | 9.2 | 16.1 | 1675 |
| 24 months + | 81.4 | 7.3 | 11.3 | 18.6 | 5172 |
| **Sex of child*** | | | | | |
| Male | 80.7 | 7.1 | 12.2 | 19.3 | 4849 |
| Female | 82.9 | 7.9 | 9.2 | 17.1 | 4572 |
| **Birth order*** | | | | | |
| 1–3 | 82.5 | 7.1 | 10.5 | 17.5 | 5852 |
| 4+ | 80.5 | 8.2 | 11.2 | 19.5 | 3468 |
| **Maternal BMI*** | | | | | |
| Underweight | 84.7 | 9.9 | 5.4 | 15.3 | 202 |
| Normal Weight | 82.7 | 8.5 | 8.8 | 17.3 | 2963 |
| Overweight/Obese | 79.5 | 5.8 | 14.7 | 20.5 | 3212 |
| **Maternal age at delivery*** | | | | | |
| Below 20 years | 84.2 | 9.7 | 6 | 15.8 | 349 |
| 20-34years | 81.8 | 7.5 | 10.7 | 18.2 | 7514 |
| 35-49years | 81.1 | 7 | 11.9 | 18.9 | 1559 |
| **Maternal Education*** | | | | | |
| No formal | 75.2 | 18.6 | 6.2 | 24.8 | 420 |
| Primary | 82.4 | 9.3 | 8.3 | 17.6 | 1021 |
| Secondary and higher | 82 | 6.7 | 11.3 | 18 | 7981 |
| **Wealth index*** | | | | | |
| Poor | 79 | 10.5 | 10.5 | 21 | 276 |
| Middle | 80.1 | 11 | 8.9 | 19.9 | 853 |
| Rich | 82 | 7 | 10.9 | 18 | 8293 |
| **ANC Registration*** | | | | | |
| Early registration | 81.1 | 6 | 12.9 | 18.9 | 2200 |
| Late registration | 82.2 | 8.2 | 9.6 | 17.8 | 4365 |
| **ANC visits*** | | | | | |
| Less than 4 visits | 78.6 | 12.2 | 9.2 | 21.4 | 468 |
| 4 or more visits | 82.1 | 7.2 | 10.7 | 17.9 | 5871 |
| **Skilled antenatal care** | | | | | |
| No | 76.2 | 8.1 | 15.7 | 23.8 | 185 |
| Yes | 81.8 | 7.5 | 10.7 | 18.2 | 6458 |
| **Year of births*** | | | | | |
| 2008–2013 | 79.8 | 7.6 | 12.6 | 20.2 | 3818 |

*(Continued)*

**Table 1.** (Continued)

| Characteristics | Normal (NBW) | Abnormal Birth weight (ABW) | | | N |
| --- | --- | --- | --- | --- | --- |
| | | Low (LBW) | High (HBW) | LBW & HBW | |
| 2013–2018 | 83 | 7.5 | 9.5 | 17 | 5604 |
| **Overall** | 81.7 | 7.5 | 10.7 | 18.3 | 9422 |

* Chi-square test of association significant at p<0.05.

pattern of HBW changed in model 2 as North West (OR 0.22; 95% CI 0.10–0.48) remained the only region that was significantly different from the South West. Infants in the South West were about five times more likely to be HBW than the North West. Mothers in the South East (OR 1.11; 95% CI 0.83–1.49) and South South (OR 1.31; 95% CI 0.81–1.88) reported the highest odds of HBW. Maternal BMI, sex of the child, maternal education and ANC registration were significant predictors of HBW in urban Nigeria. Overweight/obese mothers were 1.6 times more likely to have HBW than mothers with normal weight (OR 1.66; 95% CI 1.29–2.12). Male infants were at higher odds of HBW (OR 1.43; 95% CI 1.41–1.81). Women with primary education had the lowest odds (OR 0.64; 95% CI 0.43–1.94) of having macrosomia infants than other education categories. Also, delayed onset of ANC (OR 0.76; 95% CI 0.60–0.96) increased the risk of HBW by 24%.

## Discussion

This is the first study that examined abnormal birth weights in urban Nigeria using NDHS pool dataset. The findings show that the prevalence of ABW in urban Nigeria was 18.3 percent, which implies that at least one out of six livebirths in urban Nigeria for the study period was either low weight at birth infant or macrosomia infant. High birth weight accounted for the majority of infants who are outside the normal birth weight range values of 2,500g to 4,000g in urban Nigeria. This is a high prevalence of HBW (10.7 percent) than LBW (7.5 percent) in urban Nigeria. Community, maternal and child characteristics as well as healthcare utilization are significantly associated with both categories of abnormal birth weight- LBW and HBW- in urban Nigeria.

This pattern of a higher proportion of HBW among abnormal birth weights is in contrast to findings in similar African countries- Ghana and Ethiopia [28–30] where LBW was higher than HBW. However, these studies were hospital-based and included both urban and rural areas, which may account for some of the differences. Another study [24] on ABW among full term singletons in two urban provinces in a high-income country, reported higher HBW of 17.1 percent to 1.6 percent LBW, though consistent with this research, has a higher estimate.

The literature suggests that common factors that affect LBW and HBW are maternal and child characteristics, healthcare utilization and community variables [14, 19, 23, 26–36, 40, 47, 51]. These risk factors for low birth weight have been identified in both hospital-based study [46] and national surveys [45] in Nigeria. The factors which are likewise significant for high birth weights, though not comprehensive as in the previous studies [23, 28, 30], have been reported globally as well as in Nigeria [26, 27, 29, 40, 47].

Although this result is based on data from urban areas of Nigeria, a study [45] that used 2013 NDHS dataset for both rural and urban areas of Nigeria, confirms geopolitical region as a significant predictor of abnormal birth weights among others. Infants from the northern region have higher odds of LBW in urban Nigeria while those from southern part have a higher prevalence of HBW.

**Table 2. Adjusted odds ratio and 95% confidence interval for predictors of LBW and HBW in urban Nigeria.**

| Characteristics | LBW | | HBW | |
|---|---|---|---|---|
| | Model 1 | Model 2 | Model 1 | Model 2 |
| **Region** | | | | |
| North West | 3.91 (3.15–4.86)* | 6.15 (4.15–9.11)* | 0.26 (0.18–0.40)* | 0.22 (0.10–0.48)* |
| North East | 2.54 (1.89–3.39)* | 2.80(1.57–5.00)* | 0.71 (0.50–0.99)* | 0.74 (0.40–1.36) |
| North Central | 1.35 (1.01–1.80)* | 1.02 (0.54–1.92) | 0.93 (0.74–1.19) | 0.98 (0.64–1.46) |
| South East | 0.79 (0.62–1.01) | 0.58 (0.34–1.00) | 1.19 (1.02–1.40)* | 1.11 (0.83–1.49) |
| South South | 1.36 (1.04–1.78)* | 1.30 (0.75–2.26) | 1.41 (1.16–1.72)* | 1.31 (0.91–1.88) |
| South West (ref.) | 1 | 1 | 1 | 1 |
| **Type of Birth** | | | | |
| Singleton | | 0.19 (0.10–0.35)* | | 0.90 (0.43–1.89) |
| Multiple (ref.) | | 1 | | 1 |
| **Birth Interval** | | | | |
| Less than 24 months | | 0.98 (0.66–1.45) | | 0.86 (0.64–1.15) |
| 24 months + (ref.) | | 1 | | 1 |
| **Birth order** | | | | |
| 1–3 | | 0.72 (0.52–0.99)* | | 0.89 (0.69–1.14) |
| 4+ (ref.) | | 1 | | 1 |
| **Sex of child** | | | | |
| Male | | 1.07 (0.79–1.44) | | 1.43 (1.14–1.81)* |
| Female (ref.) | | 1 | | 1 |
| **Maternal BMI** | | | | |
| Underweight | | 0.32 (0.09–1.18) | | 0.59 (0.22–1.57) |
| Overweight/Obese | | 0.68 (0.49–0.92)* | | 1.66 (1.29–2.12)* |
| Normal Weight (ref.) | | 1 | | 1 |
| **Maternal age at delivery** | | | | |
| Below 35 years | | 1.09 (0.75–1.59) | | 0.84 (0.65–1.10) |
| 35-49yrs (ref.) | | 1 | | 1 |
| **Maternal Education** | | | | |
| No formal | | 1.25 (0.69–2.24) | | 1.37 (0.73–2.58) |
| Primary | | 1.07 (0.70–1.64) | | 0.64 (0.43–0.94)* |
| Secondary & Higher(ref.) | | 1 | | 1 |
| **Wealth index** | | | | |
| Poor | | 0.76 (0.26–2.18) | | 1.27 (0.58–2.77) |
| Middle | | 0.83 (0.47–1.45) | | 1.09 (0.70–1.70) |
| Rich (ref.) | | 1 | | 1 |
| **ANC Registration** | | | | |
| Early registration | | 1.32 (0.91–1.92) | | 0.76 (0.60–0.96)* |
| Late registration (ref.) | | 1 | | 1 |
| **ANC visits** | | | | |
| Less than 4 visits | | 0.94 (0.49–1.79) | | 0.81 (0.43–1.52) |
| 4 or more visits (ref.) | | 1 | | 1 |
| **ANC care by SHW** | | | | |
| Non-Skilled HW | | 1.07 (0.33–3.45) | | 1.50 (0.63–3.60) |
| Skilled HW (ref.) | | 1 | | 1 |

* Significant at p<0.05.

The regional differences in the reported pattern of ABW in urban areas conform to the expected north-south dichotomy in health indicators and outcomes in Nigeria [48, 49]. The differences can be linked to variations in the adoption of western education, urbanization rate and pace over time, as well as the broad tendencies and geographical differences in the dynamics of the various cultural, social and economic factors such as conflict and displacement all contributing to food insecurity [9]. For instance, the North West and North East regions that have the highest estimates of LBW also have the highest total fertility rates (NW:6.6 NE:6.1) [48], stunting rates (NW:50.4%, NE:42.8%) [52] and under-five mortality rates (NW:187, NE:134) [48]. This may partially explain the high prevalence rates of acute malnutrition among women and severe underweight among children under five years old in these two regions of Nigeria [48, 52]. The two regions are probably facing some of the foremost emerging issues of concern in the urbanization of sub-Saharan Africa.

On the other hand, the southern regions have rapidly adapted urban advantage of education and urban lifestyle, with its adverse consequence. As suggested by another study [53], one of the negative effects of adapting urbanized and westernized lifestyle is the prevalence of over-nutrition. This can have an adverse effect on maternal quality of life during pregnancy because of unhealthy food consumption [54] and obesity, a health problem which is on the increase with gestational diabetes. This study established that overweight and obese mothers have more macrosomia infants, with a higher proportion of such infant in the southern part of urban Nigeria.

Moreover, the regional dichotomy in LBW and HBW is supported by the estimates on nutritional status of women of reproductive age which showed the variation in the burden of undernutrition and over nutrition in northern and southern Nigeria. The recent 2019 Nigeria Demographic and Health Survey [48] reported proportion of women whose BMI are less than 18.5 to be 9.2% in North Central, 23.3% in North East, 16.9% in North West, 5.0% in South East, 5.7% in South South and 8.5% in South West. In contrast, women whose BMI are 25.0 and higher are 25.8% in North Central, 15.1% in North East, 16.4% in North West, 39.5% in South East, 42.9% in South South, and 37.8% in South West.

Maternal BMI, an anthropometric measure of maternal nutrition, can affect the growth and development of the fetus. An underweight mother is likely to be malnourished in some important nutritional components during pregnancy and can lead to LBW while overweight mothers have macrosomia babies. Some previous studies [14, 19] support the assertion that underweight mothers have a higher prevalence of LBW than other maternal BMI groups. Also, the fact that overweight/obese mothers were more likely to have HBW infants than normal weight mothers substantiates an assertion [25] that increase in maternal anthropometry has led to an increase in the weight of infants born at or after term.

Furthermore, studies [19, 33] have established that antenatal health care utilization is important as the pathways through which healthcare use impacts birth weight through nutritional counselling and foetus monitoring during antenatal visits. It was however, only significant for abnormal birth weights categories in this study of urban residents at the bivariable level for timing of registration and number of visits.

This study has some limitations. Maternal BMI is used as a proxy for the maternal anthropometric parameter at birth because the information collected during the NDHS may not be completely applicable at the birth of the index child, which may be within five years preceding the survey. While maternal weight and nutrition have been found as a significant risk factor to birth weight, other factors such as maternal education and household wealth index considered in this study are likely to predict maternal nutritional status and anthropometric measures in pregnancy. Another limitation of the study is the non-distinction of full-term or preterm birth in the NDHS dataset since preterm babies are known to have lower birth weight, and general

low reporting of birth weight in Nigeria with estimates of 16 percent and 24 percent reported birth weight in NDHS 2013 and 2018 respectively. The study also recognized the limitation of non-inclusion of other factors like maternal infections that can contribute to preterm birth and LBW, which were not captured in the Nigerian Demographic and Health Survey.

## Conclusion

Nigeria is a pronatalist society and it is important to note that abnormal birth weight is high in urban Nigeria. This study also provides empirical evidence on the prevalence of macrosomia from analysis of a nationally representative survey data that covers ten years in urban Nigeria. While previous focus had been more on the reduction of LBW, the findings from this study provide a basis to incorporate measures that can reduce HBW as well as improve child health outcome in the context of rapid urbanization. Interventions on ABW in urban Nigeria should be designed around community and socioeconomic indicators. Future studies can explore the rural-urban differentials in high birth weight in Nigeria.

## Author Contributions

**Conceptualization:** Olufunke Fayehun, Soladoye Asa.

**Formal analysis:** Olufunke Fayehun.

**Funding acquisition:** Olufunke Fayehun.

**Methodology:** Olufunke Fayehun.

**Writing – original draft:** Olufunke Fayehun.

**Writing – review & editing:** Olufunke Fayehun, Soladoye Asa.

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
