## [Decision Letter · Decision Letter 0]

29 Oct 2020

PONE-D-20-16572

Abnormal birth weight in urban Nigeria: an examination of related factors

PLOS ONE

Dear Dr. Fayehun,

Thank you for submitting your manuscript to PLOS ONE. After careful consideration, we feel that it has merit but does not fully meet PLOS ONE’s publication criteria as it currently stands. Therefore, we invite you to submit a revised version of the manuscript that addresses the points raised during the review process.

The reviewer has recommended major revisions to your article. Kindly pay attention to concerns around data interpretation.

We look forward to receiving your revised manuscript.

Kind regards,

Tanya Doherty, PhD

Academic Editor

PLOS ONE

Journal Requirements:

Reviewers' comments:

Reviewer's Responses to Questions

**Comments to the Author**

1. Is the manuscript technically sound, and do the data support the conclusions?

Reviewer #1: Partly

Reviewer #2: Yes

2. Has the statistical analysis been performed appropriately and rigorously? 

Reviewer #1: Yes

Reviewer #2: Yes

3. Have the authors made all data underlying the findings in their manuscript fully available?

Reviewer #1: Yes

Reviewer #2: Yes

4. Is the manuscript presented in an intelligible fashion and written in standard English?

Reviewer #1: Yes

Reviewer #2: Yes

5. Review Comments to the Author

Reviewer #1: The authors have used national datasets to explore an important issue of abnormal birthweight – considering not only small babies but also large babies in the context of urbanization. The findings show a higher proportion of macrosomia in urban Nigeria, which does not get as much attention as LBW in sub-Saharan Africa.

Methods

• Clarity in your methods that you modeled odds of LBW and HBW separately and not ABW pooled, which is how it reads.

Results

• In the results (i.e., results paragraph 5) you do not need to repeat your methods and just present the results.

• Results on Table 1: remove references to more LBW and more HBW among those who did not receive ANC from skilled providers – your results show there was no significant difference in birth weight based on this variable.

• The results text could be abbreviated to the main points. Paragraphs 1-4 referring to table 1 would be better as one paragraph and summarized to the main highlights.

• Results paragraph 6: The inclusion of individual and health care utilization didn’t decrease the odds of regional variation but the odds ratios were no longer significant when accounting for individual and healthcare variables, except in the North West and North East. Be careful with the language since “X decreases Y” implies a causal relationship. Your description in Paragraph 7 for HBW is much better.

• Results should include the numerical results, you sometimes present them and sometimes do not. Please be succinct and present the main findings and numerical results together (i.e. All included variables for LBW in Model 2 were not significant except lower odds among singleton births (OR 0.19, 95% CI 1.0-0.35, etc.). But also check the results, it appears the 95% CI for LBW Model 2 type of birth has a typo. Paragraph 7 reporting HBW models should also be revised to follow this approach.

Discussion

• The discussion could use some better organization. It was a bit hard to follow the main points you were trying to make. Maybe start the first paragraph of the discussion to highlight your main findings and then unpack each main finding in one paragraph.

• For your first finding… It seemed to me that you are mainly saying: HBW were a bigger concern than LBW in urban Nigeria. You are right in saying we focus much more on LBW in sub-Saharan Africa but with urbanization we need to also be focusing on the issue of overnutrition and the negative implications for newborns. It would be helpful to have some data to support this on the burden of overnutrition in Nigeria to help support this claim for readers who are not so familiar with the data in urban Nigeria.

• Please also make sure each time you refer to studies (i.e., Canada study mentioned in discussion paragraph 1), that you always put the reference.

• There are other factors that can contribute to preterm birth and LBW that are importantly missing from your study (because they are not part of DHS datasets), like maternal infections.

• Shorter birth intervals are often associated with preterm (and thus often LBW) births but your study didn’t find that. Any thoughts why?

• In the paragraph on the region, you talk about the north having lower health statistics in general but many that you are talking about, such as stunting are also directly related to one of your outcomes of LBW. What factors, perhaps not measured in your study, may be accounting for these types of poor health outcomes in the North – instability? Food insecurity?

• You mention in the limitations that you don’t have data on preterm births. That is fine, but Nigeria has a relatively high rate of preterm birth which may be why you are seeing factors we would expect to have an association with LBW – such as maternal undernutrition – not being significant in your study if Nigeria’s low weight births are driven by prematurity rather than intrauterine growth restriction.

• Limitations – should also likely include quality of the birthweight data. In your intro you mention birthweight is often not taken and so you are relying on maternal report of birthweight or review of records when available and so if birthweight is not taken, then mothers may not have had much information to refer to when reporting.

Minor typographical issues to correct. There is no need to respond to these in your response to reviewers, but I am just noting them as small observations I had as I read the paper.

• Consistent use of commas (or not) for reporting values in the thousands in text and tables

• Consistent use of decimal when reporting results (one decimal point or rounded to the whole number)

• When reporting the statistic for significance, please include the “p” rather than just stating the value (i.e., p<0.05);

• Table 1 – the “chi” symbol is not showing up in the table note

• Use “*” to present p<0.05 in Table 1 the same way you do in Table 2 for ease to the reader

• Make sure there is a space between the numbers and the word “Percent”

Reviewer #2: The authors must be congratulated on an excellent piece of work. The study is well designed and conducted. The methodology is comprehensive, the results clearly presented, the discussion is relevant and the conclusions and limitations appropriate. The manuscript is very well written and adheres to both scientific and language norms.

6. PLOS authors have the option to publish the peer review history of their article (what does this mean?). If published, this will include your full peer review and any attached files.

Reviewer #1: No

Reviewer #2: **Yes: **Daynia Ballot

---

## [Author Response · Author response to Decision Letter 0]

7 Nov 2020

Response to Reviewers' comment

Methods

• Clarity in your methods that you modeled odds of LBW and HBW separately and not ABW pooled, which is how it reads. 

Response: Done. Page 7

Results

• In the results (i.e., results paragraph 5) you do not need to repeat your methods and just present the results.

Response: Done. We deleted the repetition

• Results on Table 1: remove references to more LBW and more HBW among those who did not receive ANC from skilled providers – your results show there was no significant difference in birth weight based on this variable.

Response: Done. We removed the statement.

• The results text could be abbreviated to the main points. Paragraphs 1-4 referring to table 1 would be better as one paragraph and summarized to the main highlights. 

Response: Done. Thank you. We summarized the main highlight to one paragraph as suggested.

• Results paragraph 6: The inclusion of individual and health care utilization didn’t decrease the odds of regional variation but the odds ratios were no longer significant when accounting for individual and healthcare variables, except in the North West and North East. Be careful with the language since “X decreases Y” implies a causal relationship. Your description in Paragraph 7 for HBW is much better.

Response: Done. We rephrased the sentence. Thank you.

• Results should include the numerical results, you sometimes present them and sometimes do not. Please be succinct and present the main findings and numerical results together (i.e. All included variables for LBW in Model 2 were not significant except lower odds among singleton births (OR 0.19, 95% CI 1.0-0.35, etc.). But also check the results, it appears the 95% CI for LBW Model 2 type of birth has a typo. Paragraph 7 reporting HBW models should also be revised to follow this approach. 

Response: Done. We corrected the singleton birth estimate typo. Estimates included in the paragraphs. Thank you

Discussion

• The discussion could use some better organization. It was a bit hard to follow the main points you were trying to make. Maybe start the first paragraph of the discussion to highlight your main findings and then unpack each main finding in one paragraph. 

Response: Done. We revised this section. We highlighted the main findings in the first paragraph and discussed the significance of these findings in the subsequent paragraphs 

• For your first finding… It seemed to me that you are mainly saying: HBW were a bigger concern than LBW in urban Nigeria. You are right in saying we focus much more on LBW in sub-Saharan Africa but with urbanization we need to also be focusing on the issue of overnutrition and the negative implications for newborns. It would be helpful to have some data to support this on the burden of overnutrition in Nigeria to help support this claim for readers who are not so familiar with the data in urban Nigeria. 

Response: Done. We included some data to show the burden of over nutrition and undernutrition in southern and northern part of Nigeria respectively.

• Please also make sure each time you refer to studies (i.e., Canada study mentioned in discussion paragraph 1), that you always put the reference. 

Response: Done. Reference inserted

• There are other factors that can contribute to preterm birth and LBW that are importantly missing from your study (because they are not part of DHS datasets), like maternal infections. 

Response: Done. We included this as part of the study limitation

• Shorter birth intervals are often associated with preterm (and thus often LBW) births but your study didn’t find that. Any thoughts why?

Response: This may be due to the quality of data set we have. As noted in the limitation, there is a general low reporting of birth weights in Nigeria.

• In the paragraph on the region, you talk about the north having lower health statistics in general but many that you are talking about, such as stunting are also directly related to one of your outcomes of LBW. What factors, perhaps not measured in your study, may be accounting for these types of poor health outcomes in the North – instability? Food insecurity? 

Response: As noted in the discussion, there is variations in the adoption of western education, urbanization rates and pace over time because of broad tendencies and the geographical difference in the dynamics of the various cultural, social and economic factors. 

• You mention in the limitations that you don’t have data on preterm births. That is fine, but Nigeria has a relatively high rate of preterm birth which may be why you are seeing factors we would expect to have an association with LBW – such as maternal undernutrition – not being significant in your study if Nigeria’s low weight births are driven by prematurity rather than intrauterine growth restriction. 

Responses: Noted. Because of lack of data, we cannot specifically say this is the situation among the study population of urban residents in Nigeria.

• Limitations – should also likely include quality of the birthweight data. In your intro you mention birthweight is often not taken and so you are relying on maternal report of birthweight or review of records when available and so if birthweight is not taken, then mothers may not have had much information to refer to when reporting. 

Responses: Done. We have included limitations about the quality of birthweight data

Minor typographical issues to correct. There is no need to respond to these in your response to reviewers, but I am just noting them as small observations I had as I read the paper. 

• Consistent use of commas (or not) for reporting values in the thousands in text and tables

• Consistent use of decimal when reporting results (one decimal point or rounded to the whole number)

Response: Corrected in the result section

• When reporting the statistic for significance, please include the “p” rather than just stating the value (i.e., p<0.05); 

Response: Corrected in the result section

• Table 1 – the “chi” symbol is not showing up in the table note

• Use “*” to present p<0.05 in Table 1 the same way you do in Table 2 for ease to the reader

Response: Corrected in Table 1

• Make sure there is a space between the numbers and the word “Percent”

Response: Corrected.

---

## [Editor Report · Decision Letter 1]

10 Nov 2020

Abnormal birth weight in urban Nigeria: an examination of related factors

PONE-D-20-16572R1

Dear Dr. Fayehun,

We’re pleased to inform you that your manuscript has been judged scientifically suitable for publication and will be formally accepted for publication once it meets all outstanding technical requirements.

Kind regards,

Tanya Doherty, PhD

Academic Editor

PLOS ONE

Additional Editor Comments (optional):

Thank you for addressing the reviewer comments. Your paper is now deemed suitable for publication.
---

## [Editor Report · Acceptance letter]

12 Nov 2020

PONE-D-20-16572R1 

Abnormal birth weight in urban Nigeria: an examination of related factors 

Dear Dr. Fayehun:

I'm pleased to inform you that your manuscript has been deemed suitable for publication in PLOS ONE. Congratulations! Your manuscript is now with our production department. 

Kind regards, 

on behalf of

Professor Tanya Doherty 

Academic Editor

PLOS ONE